# Effect of Vitamin C on Tendinopathy Recovery: A Scoping Review

**DOI:** 10.3390/nu14132663

**Published:** 2022-06-27

**Authors:** David C. Noriega-González, Franchek Drobnic, Alberto Caballero-García, Enrique Roche, Daniel Perez-Valdecantos, Alfredo Córdova

**Affiliations:** 1Department of Surgery, Ophthalmology, Otorhinolaryngology and Physiotherapy, Faculty of Medicine, Hospital Clínico Universitario de Valladolid, 47002 Valladolid, Spain; davidcesar.noriega@uva.es; 2Medical Services Shanghai Shenhua FC, Shanghai 201315, China; docdrobnic@gmail.com; 3Department of Anatomy and Radiology, Faculty of Health Sciences, GIR Physical Exercise and Aging, Campus Los Pajaritos, University of Valladolid, 42004 Soria, Spain; alberto.caballero@uva.es; 4Department of Applied Biology-Nutrition, Institute of Bioengineering, University Miguel Hernández, 03202 Elche, Spain; eroche@umh.es; 5Alicante Institute for Health and Biomedical Research (ISABIAL), 03010 Alicante, Spain; 6CIBER Fisiopatología de la Obesidad y Nutrición (CIBEROBN), Instituto de Salud Carlos III (ISCIII), 28029 Madrid, Spain; 7Department of Biochemistry, Molecular Biology and Physiology, Faculty of Health Sciences, GIR Physical Exercise and Aging, Campus Duques de Soria, University of Valladolid, 42004 Soria, Spain; danielperezvaldecantos@gmail.com

**Keywords:** vitamin C, ascorbic acid, tendinopathy, collagen, oxidative stress

## Abstract

Tendinopathies represent 30–50% of all sports injuries. The tendon response is influenced by the load (volume, intensity, and frequency) that the tendon support, resulting in irritability and pain, among others. The main molecular component of tendons is collagen I (60–85%). The rest consist of glycosaminoglycans-proteoglycans, glycoproteins, and other collagen subtypes. This study’s aim was to critically evaluate the efficacy of vitamin C supplementation in the treatment of tendinopathies. At the same time, the study aims to determine the optimal conditions (dose and time) for vitamin C supplementation. A structured search was carried out in the SCOPUS, Medline (PubMed), and Web of Science (WOS) databases. The inclusion criteria took into account studies describing optimal tendon recovery when using vitamin C alone or in combination with other compounds. The study design was considered, including randomized, double-blind controlled, and parallel designs in animal models or humans. The main outcome is that vitamin C supplementation is potentially useful as a therapeutic approach for tendinopathy recovery. Vitamin C supplementation, alone or in combination with other products, increases collagen synthesis with a consequent improvement in the patient’s condition. On the other hand, vitamin C deficiency is mainly associated with a decrease in procollagen synthesis and reduced hydroxylation of proline and lysine residues, hindering the tendon repair process.

## 1. Introduction

It has been described that vitamin C (VC) is important in tendon and ligament healing, mainly due to its antioxidant properties and its function as a cofactor for collagen synthesis [1,2,3,4]. As an antioxidant, VC increases intracellular levels of reduced glutathione (GSH), a main intracellular antioxidant. As a cofactor in collagen synthesis, VC participates in the hydroxylation of proline and lysine residues to hydroxyproline and hydroxylysine, respectively, in collagen molecules. This structural modification is instrumental for optimal mechanical tendon properties [2,5,6].

In addition, in vitro, tenocyte cultures in the presence of ascorbate produce an acceleration of intracellular procollagen secretion. In fact, ascorbate-treated tenocytes maintained a large procollagen reserve [7]. Furthermore, ascorbate treatment significantly improved tendon resistance after 6 weeks of treatment in an animal model with tendinopathy. In this context, VC-treated animals tended to reduce fibrotic scarring at the site of injury [8].

Regarding humans, tendinopathy, mainly produced by tendon overuse, is a pathology that causes pain, joint dysfunction, and reduced exercise tolerance. It is a common pathology in sports people and has been estimated about 30–50% of all sports injuries. Tendinopathy is the general term used for both tendinitis and tendinosis. While tendinitis implies inflammation of the tendon, tendinosis indicates a damaged tendon due to different problems in the tendon tissue itself [9,10].

Overuse tendinopathy, i.e., repetitive storage and release of energy and excessive compression, is problematic to manage clinically. The tendon response is influenced by the load (volume, intensity, and frequency) that the tendon supports and is associated with irritability and pain, among other symptoms [11]. In addition, other extrinsic factors such as smoking, age, and the use of certain drugs (corticosteroids or fluoroquinolones) are considered to be predisposing factors for tendinosis pathologies [12]. In consequence, the subsequent recovery is also variable. Other risk factors to consider include the presence of an inflammatory process [13], i.e., not directly related to rheumatic pathology per se, but maintaining a low grade of systemic inflammation, similar to those observed in allergic conditions [14] or impaired insulin sensitivity in obesity [15,16]. A modest inflammatory process seems to be sufficient to act negatively on the tendon structure and function, predisposing to the development of chronic overuse tendinopathies or acute tendon injury [17].

### 1.1. Tendon Structure and Homeostasis

The fundamental molecular component of tendons is type I collagen, corresponding to 60–85% of tendon dry weight. The rest are glycosaminoglycan-proteoglycans, glycoproteins, and other collagen subtypes. The cellular component are tenocytes [18,19]. The main function of tenocytes is the control intracellular metabolism, as well as extracellular actions such as formation and degradation of the extracellular matrix (ECM). All this structure aims to respond to mechanical stimuli supported by the tendon [20].

The tendon ECM turnover is influenced by physical activity, blood flow, oxygen demand, and the amount of collagen synthesized responding to collagen balance. In this line, the activity of matrix metalloproteinases (MMPs) increases with mechanical loading. On the other hand, MMP inactivity decreases collagen turnover. Therefore, human tendon tissue is highly responsive to mechanical stimuli that involve metabolic and circulatory changes as well as ECM remodeling [21].

The adaptive response of the tendon involves the participation of many inter- and intracellular signaling pathways. Targets of these pathways include the transcription factor NF-кB (nuclear factor kappa-B), ERK (extracellular signal-regulated kinases), MAPK (mitogen-activated protein kinase), and the TGF-β (transforming growth factor-β) [22].

ECM changes observed during tendinopathy are characterized by a loss of structural organization of collagen accompanied by alterations in fibrocartilaginous composition with deposition of additional matrix proteins (i.e., glycosaminoglycans). In addition, type III collagen is produced during the initial phase of tendon damage, acting as a patch to protect the damaged area. Over time, when the tendon is recovering the original structure, type I collagen replaces type III. This final biosynthetic process resumes the linear structured arrangement culminating in tendon recovery [23,24].

### 1.2. Etiopathogenesis

A model of tendon pathology and response to treatment has been considered for therapeutic interventions [9,25]. According to this model, tendon pathology has three stages: (a) reactive tendinopathy, (b) tendon destruction, and (c) degenerative tendinopathy. This model divides tendinopathies into three groups according to the main clinical factor: the tear or disruption of collagen, the inflammatory process, and the cellular response of the tendon [9,25].

#### 1.2.1. Reactive Tendinopathy (Stage I)

Usually is a consequence of acute overload due to tension or compression on the tendon. The return to physical activity after a resting period as the sole treatment may be considered in this line. Microscopically, reactive tendinopathy shows cellular hyperactivity, an increase in the number of tendon cells, and water attraction in the tissue by hydrophilic molecules (i.e., proteoglycans). No changes are observed in the tendon fibers, only swelling and thickening of the diameter [10,26,27].

#### 1.2.2. Unstructured Tendon (Stage II)

This stage appears when the recovery or adaptation process fails. This situation is characterized by the presence of greater disorganization of the cellular matrix than in reactive tendinopathy (stage I). In this phase II, there is also a loss of tendon fiber structure, accompanied by an increase in the number of cells. A significant increase in the production of collagen and proteoglycans (hydrophilic substances) is observed as well. The unstructured tendon is asymptomatic, painless, and it only would be recognizable using diagnostic imaging (ultrasounds or magnetic resonance imaging) [25].

#### 1.2.3. Degenerative Tendinopathy (Stage III)

The degenerative tendinopathy is characterized by progressive disorganization of the tendon extracellular matrix and collagen. The cells are damaged or die and present changes in blood vessels of the tendon (neovascularization). In addition, the affected areas do not show a characteristic aligned fibrillar structure, being unable to support any tensional stress. Otherwise said, these unstructured areas are believed to be mechanically very weak. The damaged tissue produces different pain messengers and activates peripheral nerves leading to hypersensitivity [25].

### 1.3. Pathophysiology

The pathogenesis of tendinopathy is multifactorial and complex. The pathological process appears to be initiated by repetitive tendon overloading, leading to structural damage of collagen fibrils. These initial alterations are usually asymptomatic. Progressive accumulation of extracellular matrix damage and the secretion of cytokines, chemokines, inflammatory mediators, and activated nociceptors eventually lead to the manifestation of symptoms [22].

Therefore, the cellular process includes inflammation, neuronal and vascular changes, and apoptosis. In addition, individual variables must be considered in the development of tendinopathies, such as genetic profile, age, gender, flexibility, diseases, and previous injuries [28].

### 1.4. Oxidative Stress

Degenerated tendons are a source of reactive oxygen species (ROS), contributing significantly to tendinopathy progression [29]. Malondialdehyde (MDA) is an important indicator of lipid peroxidation, increasing after collagenase injection into rat tendons [30]. As well as MDA, other oxidative molecules contribute to tendinopathy formation by affecting tendon homeostasis and producing inflammatory and apoptotic processes. Therefore, injuries can lead to overproduction of ROS and oxidative stress processes in the injured area. Probably, redox regulation plays a role in the healing of tendons that suffer degenerative injuries. For this reason, it seems that the antioxidant defense is instrumental for tendon healing [31,32,33].

## 2. Objectives

Actually, several compounds are used as oral treatments for tendinopathies, including glucosamine, chondroitin sulfate, VC, hydrolyzed type I collagen, L-arginine α-ketoglutarate, and curcumin, among others. These supplements seem to act to preserve or even repair damaged tendons [5,7,34,35,36,37,38,39].

In this context, VC has a particular interest due to its antioxidant properties (increasing reduced glutathione: GSH) and, at the same time, acting as a cofactor in collagen synthesis via hydroxylation of proline and lysine residues, leading to hydroxyproline and hydroxylysine respectively [2,5,6].

The main aim of this study was to critically evaluate the efficacy of VC supplementation in the treatment of tendinopathies. Moreover, the study aims to demonstrate firstly whether VC is effective and secondly what are the optimal conditions (dosage and time) in the therapeutic approach.

## 3. Materials and Methods

This review is focused on the analysis of VC supplementation in the treatment of tendinopathies. The PICOS question model was used to develop the search and define the inclusion criteria [37].

The model proposed for this review, as indicated in the title, is a scoping review. Scoping reviews answer broad research questions while maintaining the methodological rigor of systematic reviews. Its main objective is to identify and map the available evidence for a specific area.

In this context, a bibliographic search was conducted following the guidelines for trusted systematic reviews: a new edition of the Cochrane Handbook for Systematic Reviews of Interventions [40,41]. The methodological approach used to carry out this scoping review is the PRISMA (Preferred Reporting Items for Systematic Reviews and Meta-Analyses) extension for scoping reviews. [42]. The Cochrane risk-of-bias tool (Review manager version 5.4, Cochrane Collaboration, Chichester, UK) [43] was used to evaluate the quality assessment of selected studies, including domains of random sequence generation, allocation concealment, blinding of participants and personnel, blinding of outcome assessment, incomplete outcome data, and selective reporting (Figure 1). Other biases, including study design rationality and compliance with treatment, were also assessed. We rated studies that satisfy four or more of seven low-risk domains of bias as low risk, with the rest as high risk. Two investigators (F.D. and D.C.N.-G.) evaluated the risk of biases independently, with any discrepancies adjudicated by a third researcher (A.C.). Review is not registered in PROSPERO.

A structured search was conducted in the SCOPUS, Medline (Pub-Med), and Web of Science (WOS) databases. The last consultation of these databases was done in March 2022. The search used the following keywords “Vitamin C or Ascorbic acid” and “Connective tissue healing or Tendon or Tendinopathy.” All found titles and abstracts were separated to identify duplicates and possible missing studies. The suitability of the articles was determined using the GRADE concept [44] also using the level of evidence criterion [40]. All articles analyzed had a GRADE “Moderate” or “High scientific quality” and a degree of evidence that can be classified from 2 to 2++ were selected. The inclusion criteria for this review are indicated in Table 1. The “Full search strategy” is presented in Figure 2.

**Figure 1 nutrients-14-02663-f001:**
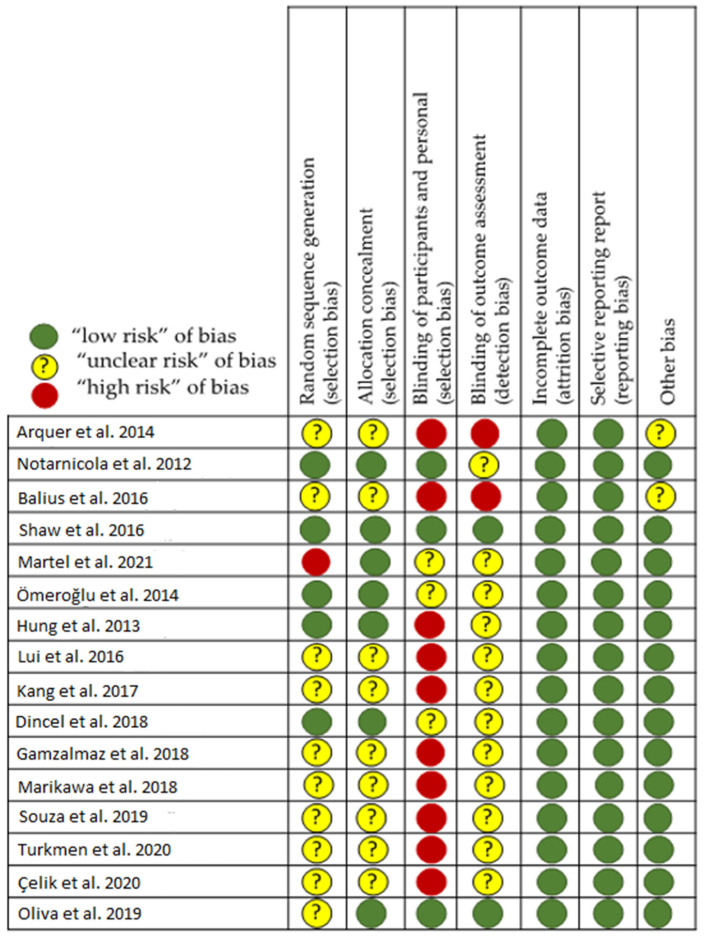
Risk of bias assessment of included studies [3,8,45,46,47,48,49,50,51,52,53,54,55,56,57,58].

## 4. Results

From the literature, 20 articles related to the select descriptors were selected, but only 16 articles met all the inclusion criteria (Figure 1 and Table 1). The 4 discarded out of the 20 were related to the use of the VC in the improvement of the healing process of the digestive surgery suture. Of the 16 remaining articles, 5 were related to humans (Table 2) and 11 to laboratory animals, being considered pre-clinical studies (Table 3).

Table 2 and Table 3 show the study reference in the first column and the variables corresponding to each study in the rest of the columns. The second column indicates whether the compound administered is associated or not with VC. In Table 2, in three studies [42,43,45], VC is administered in combination with other supposed stimulators of connective tissue formation, and in two of them, other therapeutic tools are applied [45,46,47]. In one of the studies [48], VC is added to a culture of human ligament tissue to identify its suitability for improving physical and biological properties. In another study [49], a dose of 500 mg/d is administered in relation to surgical repair of the rotator cuff. In none of the four studies where VC is administered orally, the dose exceeds this value, being in most of them 60 mg, including administration with other compounds. Moreover, the evaluation of the effect is carried out in four studies using exploratory clinical assessment and subjective criteria according to predetermined questionnaires (Analog Visual Scale-Achilles (VISA-A), Analog Visual Scale-Patella (VISA-P), Pain perception, Patient-Rated Tennis Elbow Evaluation (PRTEE), among others).

In relation to the studies on laboratory animals [3,8,50,51,52,53,54,55,56,57,58], in six of them, VC is administered alone and in the other five in combination with other products. The administration is diverse: in two studies [50,57], the tissue is treated prior to transplantation; in three [8,55,58], VC is applied to the injured area by local injection; in two [51,54], the rodent takes VC from the water according to its needs (“ad libitum”); in two studies [3,52] VC is administered by peritoneal injection and, finally, in the last two [53,56], VC is provided by gastric lavage. The last column of both tables shows the effect of the administration of the product studied according to the tests carried out.

The reparative effect is usually studied by histological, biomechanical, or imaging tests on models of tendon cutting and repair. This is the case for nine studies, while in others, recovery from a tendonitis process is studied [51] or modification of the properties of the rotator cuff [54]. One of the interesting aspects that should be evaluated and transferred from animal models to therapeutic use in humans is the oral dose administered and if VC is used alone or in combination with other compounds. Doses used range from 7.2 mg/kg [53] to 150 mg/kg [3,52,56]. The transfer to humans, considering the weight of rodents between 250 and 350 g, would correspond to 500 mg for a 70 kg subject in the first case [53] to 42–100 g in the others [3,52,56].

## 5. Discussion

First of all, and regarding the selected studies, in the assessment of bias (Figure 2), it is interesting to note that in the laboratory animal studies, randomization of animals is either absent or not discussed. It is quite possible that it is not important and does not influence the study, but it is not known. On the other hand, this means that the placement in one group or another may be compromised by some of the researchers, who do not know whether or not they are aware of the therapy administered. Finally, it is common for the pathologist to work without knowing from which animal sample comes (blind studies), but this is not indicated, except in one study [58]. In this particular study, the samples were anonymous and randomly provided to the specialist.

In general, clinical practice, VC is not commonly used to treat musculoskeletal problems. Nevertheless, the present report tries to bring some evidence that this micronutrient provided as a nutraceutical could play a role in tendon recovery post-injury.

Results from animal models suggest a role of oxidative stress in tendon degeneration. In this context, oral administration of VC was effective against rotator cuff degeneration induced by endogenous oxidative stress caused by superoxide dismutase (SOD) deficiency [54]. SOD is a key intracellular antioxidant enzyme to eliminate superoxide radicals (the first free radical produced in the electron transport chain and other cell locations). The main conclusion is that an antioxidant treatment attenuated histological changes caused by oxidative stress. Overproduction of ROS is a biochemical event observed at different stages of tissue recovery [60]. ROS interactions with collagen proteins decelerate collagen network formation and consequently delay tissue remodeling [61].

However, other studies do not support the role of VC in tendon recovery. Although a favorable effect of the VC is observed in the healing of the injured tendon in the rotator cuff, the difference was not significant in relation to the control subjects [49]. Therefore, more research is necessary to solve these contradictory results, establishing the tendon condition in which VC exerts an optimal repair process.

A point with potential interest concerns the recovery potential of VC when combined with other compounds. In this context, rats treated with a combination of VC and fibrin clot accelerate tendon healing histologically, providing a tendon with high resistance, close to a biomechanically strong tendon [57]. These facts have been confirmed by other studies [8,52], showing that both VC and hyaluronic acid have therapeutic effects on tendon healing, especially in the late phase of the tendon reparation. Other studies indicate that injection of ascorbic acid with T_3_ improves the biological function of tenocytes, favoring tendon healing [58,62].

In a nutritional context, a supplement rich in gelatin and VC (48.5 mg) increased the circulating levels of the amino acid components of collagen. Only 1 h after consuming the supplement was enough to increase the collagen content and the mechanical properties of ligaments during exercise [48]. Other studies [45] have used a food supplement based on mucopolysaccharides, type I collagen, and VC (Tendoactive^®^ (Gramm Pharmaceuticals, Grand Rapids, MI, USA): 435 mg of mucopolysaccharides, 75 mg of type I collagen, and 60 mg of VC) and studied the effect on three tendons (Achilles, patellar and lateral epicondyle tendon). The study provides evidence of the efficacy of Tendoactive^®^ (Gramm Pharmaceuticals, Grand Rapids, MI, USA) for the treatment of tendinopathies. They have observed a 69% reduction in pain in patients with Achilles tendinopathy, 83% in patients with tennis elbow, and 75% in patients with supraspinatus tendinopathy. This was associated with clear structural improvements, including a reduction in the thickness of the affected tendon. Similar results were found by supplementation with mucopolysaccharides, type I collagen, and VC combined with a passive stretching program [47].

On the other hand, VC deficiency is associated with decreased procollagen synthesis and reduced hydroxylation of proline and lysine residues [63,64]. In studies in guinea pigs, proline hydroxylation in articular cartilage has been shown to be especially resistant to ascorbate deficiency [65]. Under these conditions, collagen synthesis decreased to 50% with respect to controls. Also, in cartilage from scorbutic guinea pigs, proteoglycan synthesis decreased at the same time as collagen synthesis, and there is a direct correlation between the rates of synthesis of both compounds and the rate of weight loss [66]. Therefore, after injury, the levels of VC required for adequate healing could be higher than those required for homeostasis in a steady-state [2].

In the treatment of tendinopathies, the use of nonsteroidal anti-inflammatory drugs (NSAIDs) is common, regardless of supplementation with other molecules such as those described earlier in this review. It is important to note that NSAIDs may affect the healing of the tissues that take part in the structure of the musculoskeletal system, affecting specifically the proliferation of tenocytes after injury [67]. Therefore, the existing scientific references do not provide enough evidence for or against the use of NSAIDs after an acute injury or surgical repair of the tendon-bone interface [68,69,70]. However, the combination with supplements containing VC [71] or other compounds, such as hyaluronic acid [72], could mitigate the effects exerted by NSAIDs. Nevertheless, direct administration of NSAIDs in the peritendinous area seems to be effective for the anesthetic (reducing pain) and anti-inflammatory process [73].

Finally, various local physical therapy applications have been shown to be beneficial, such as eccentric exercises, ultrasound (US), laser therapy, and extracorporeal shockwave therapy (ESWT) [74,75,76,77,78]. The ESWT is effective for tendinopathy [79], stimulating soft-tissue healing primarily by inhibiting afferent pain-receptor function [80], downregulating the expression of inflammatory cytokines [81], improving cellular proliferation and synthesis of ECM matrix [82], and enhancing angiogenesis [83].

## 6. Conclusions

In view of the studies analyzed, it seems clear the usefulness of VC in the therapeutic approach to tendinopathies. A supplementation of VC, alone or combined with other compounds, increases the production of collagen, with the consequent improvement of recovery in the patient. It is important to consider that many of the studies have been developed for injectable administration of VC in the affected area. In addition, VC deficiency is fundamentally associated with a decrease in procollagen synthesis and reduced hydroxylation of proline and lysine residues, hindering the tendon repair process. Despite this, there is no unanimity on the more efficient doses to be used. At the moment, 60 mg of VC alone or in combination with other compounds seems to be the dose mainly proposed for tendinopathy treatment. Nevertheless, when VC is taken alone as an antioxidant, higher doses have been used. Therefore, more studies have to be carried out to determine the optimal oral dose that could be useful in the resolution of this pathology. Finally, the present report is focused only on VC, but nutrition in general and exercise have to be considered together for optimal and healthy performance.

## Figures and Tables

**Figure 2 nutrients-14-02663-f002:**
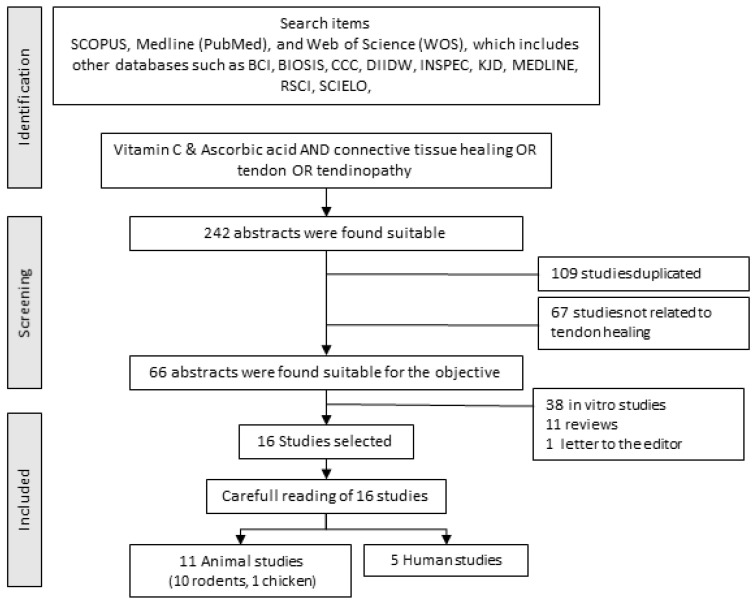
Full search strategy to develop the systematic review.

**Table 1 nutrients-14-02663-t001:** Inclusion criteria for study selection.

Main Criteria	Details
Use of vitamin C	Alone or as a supplement with other products
Therapeutic results	Restoration of tendon tissue
Methodology	Study design: randomized/double-blind controlled/parallel Animal samples Human Beings
Language	Only in English

**Table 2 nutrients-14-02663-t002:** Studies evaluating the effect of vitamin C in humans.

Reference	Compounds/Intervention	VC Dosage	Route	d	Placebo/Control	*n* (M/F)	Sample/Gender/Sport Practice	Site/Cause of Injury	Tests	Impact on Resolution	
Arquer et al. [45]	VC, collagen I, MCPg	60 mg	Oral	90	-	70 (NS)	Human	Elbow, Achilles, Knee tendinopathy	VISA-A, VISA-P, PRTEE, US	⊕ VISA-A, VISA-P, PRTEE (*p* < 0.001); ⊕ Tendon diameter (*p* < 0.05); ⊕ Pain perception	
Notarmicola et al. [46]	Arginine & combination of VC, MSM, collagen, bromelain/Shock wave T	60 mg	Oral	60 & 180	Placebo	32 (16/16)	Human		VAS, Ankle-Hindfoot Scale Roles & Maudsley score	⊕ Ankle food scale, ≈ VAS, Better PS	
Balius et al. [47]	VC, collagen I, MCPg/Passive stretching or Eccentric exercise	60 mg	Oral	84	-	59 (47/12)	Human	Achilles tendinopathy	VISA-A, US	⊕ VISA-A, Pain perception (*p* < 0.05); ⊕ Tendon diameter, Better PS with supplement	
Shaw et al. [48]	VC	48 mg VC + 5 g Gelatin	Culture media	3	Control	8 (8/0)	In vitro/Human tissue	Ligament	Histology and biomechanical tests	⊕ Biomechanical properties (*p* < 0.05);	-
		48 mg VC + 15 g Gelatin				8 (8/0)					⊕ Collagen I, (*p* < 0.05); collagen synthesis, AA
Martel et al. [49]	VC	500 mg	Oral	45	Control	98 (48/50)	Human	Rotator cuff arthroscopy repair		⊕ Tendency to a better repair. ⊕ Oximetry	

Symbols and abbreviations used: ⊕: effective; ≈: ineffective; AA: amino acids; d: days; F: females; M: males; MCPg: mucopolygel; MSM: methylsulfonylmethane; NS: non-specified; PRTEE: Patient-Rated Tennis Elbow Evaluation; PS: patient satisfaction; US: Ultrasound; VISA-A: Analog Visual Scale-Achilles; VISA-P: Analog Visual Scale-Patella; VAS: visual analog scale; VC: vitamin C.

**Table 3 nutrients-14-02663-t003:** Studies evaluating the effect of vitamin C in animal models.

Reference	Compounds	VC Dosage	Route	Days	Placebo/Control	*n*	Animal	Injury	Tests	Impact on Resolution	
Ömeroğlu et al. [3]	VC	150 mg– 1.5 mL	Peritoneal injection all days	3–10–21	Control	42	Rodents	Achilles C/R	Histology	⊕ Collagen I, fiber diameter and alignment, Better initial angiogenesis (*p* < 0.001)	
Hung et al. [8]	VC	5 mg/mL	Local injection	1	Control	22	Chicken	FxDP C/R	Histology and biomechanical tests	⊕ GSH (*p* < 0.05); ⊕ GSSG (*p* < 0.05); ⊕ Adhesion (*p* < 0.05); ⊕ Gliding resistance, Better flexor angle (*p* < 0.05)	⊕ Fibrotic size (*p* < 0.05);
		50 mg/mL				22					-
Lui et al. [50]	VC CTGF TDSC	-	Transplant pretreated connective tissue	14	Control	153	Rodents	Patellar tendon C/R	Histology, US, CT imaging, and biomechanical tests	⊕ Cellularity, ⊕ Collagen fibers alignment ⊕ Ossified depot. (*p* < 0.05)	
Kang et al. [51]	VC	1.5 g/L	“ad libitum”	28	Control	7	Rodents	Achilles tendonitis	Histology, BS	Second Better BS, ⊕ Serum VC	
	VC ASC		“ad libitum” & Local Injection (ASC)			7				⊕ BS, ⊕ Serum VC (*p* < 0.01)	
	ASC	-	Local Injection (ASC)			7				BS better than control group (*p* < 0.01)	
Dincel et al. [52]	Hyaluronic acid	0.075 mg/kg	Local injection	1	Control	16	Rodents	Achilles C/R	BS, Moving Test, Histology, and biomechanical tests	⊕ BS ⊕ Moving Test	⊕ Mean force day 15th (*p <* 0.05)
	VC	150 mg	Peritoneal alternate days	15 & 30		16					⊕ Mean forcé day 30th (*p <* 0.05)
Gemalmaz et al. [53]	VC Collagen I MCPSg	7.2 mg/kg	Gastric lavage	21	Placebo	16	Rodents	Achilles C/R	Histology and biomechanical tests	⊕ PCNA, ⊕TGF-β1 (endotendon). ⊕ biomechanical properties, collagen strength (*p <* 0.05)	
Morikawa et al. [54]	VC	1%	“ad libitum”	56	Control	56	Rodents	Rotator cuff	Histology	⊕ histologic changes	
Souza et al. [55]	VC	≈0.21 mg ^(a)^	Local injection alternate days	Until days 12 & 20	Control	6	Rodents	Achilles C/R	Histology	⊕ Achilles function index, ⊕ Collagen network	
Turkmen et al. [56]	VC Collagen I MCPg	≈0.15 mg ^(b)^	Gastric lavage		Placebo	20	Rodents	Achilles C/R	Histology and biomechanical tests	Optimal alignment in collagen fibers, ≈ Vascularization, ≈ Inflammation, ≈ biomechanical properties,	
Çelik et al. [57]	VC Fibrin clot	-	Healing clots	1	Control	20	Rodents	Achilles C/R	Histology and biomechanical tests	⊕ histologic changes, ⊕ Strength, ⊕ FGF, ⊕ VEGF (*p <* 0.05)	
Oliva et al. [58]	VC, BMSC, and T_3_, alone and in multiple combinations	≈2.5 μg ^(c)^	Local injection	1, 2 & 4	Control	24	Rodents	Achilles C/R	Histology	The combination Vitamin C + T_3_: ⊕ fiber alignment, (*p <* 0.0001),⊕ Collagen I, Collagen III (*p <* 0.05), ⊕ Cellularity and Vascularity (*p <* 0.005)	

Symbols and abbreviations used: ⊕: effective; ≈: ineffective; ASC: adipose-tissue stem cells; BMSC: bone marrow mesenchymal stem cells; BS: bone score; C/R: cut and repair; CT: Computed Tomography; CTGF: connective tissue growth factor; FGF: Fibroblast Growth Factor; FxDP: flexor digitoris profundus; GSH: reduced glutathione; GSSG: oxidized glutathione; MCPg: mucopolygel; PCNA: proliferating cell nuclear antigen; T_3_: triiodothyronine; TGF-β1: tansforming growth factor beta 1; TDSC: tendon-derived stem cells; TGF-β1: transforming growth factor-β1; US: Ultrasound; VC: vitamin C; BMSC: bone marrow stromal cell; T_3_: triiodothyronine; VEGF: vascular Endothelial Growth Factor. ^(a)^: 40 μL (30 mM) ≈ 0.21 mg VC; ^(b)^: 3 mg Retendo ≈ 0.15 mg VC; ^(c)^: 50 μg/mL ≈ 2.5 μg VC. In general, tendon injury occurs together with antioxidant depletion. This could be explained due to the presence of oxidative stress during fibrogenesis because fibroblasts can generate ROS during the phagocytic phase that occurs at the beginning of tendon injury [59]. In this line, decreased levels of GSH, one of the mains intracellular antioxidants, have been reported in a chicken model of tendon adhesion in healing [8]. GSH decrease is accompanied by increases in the oxidized form of glutathione (GSSG). Local injection of VC post-injury at different doses (5 and 50 mg/mL) in chickens reduced the degree of tendon adhesion in healing. In this report [8], 5 mg/mL of VC seems to work better than 50 after 2 weeks. In any case, VC injection was better than using saline solution. However, no significant changes were reported in GSSG after VC injection. Altogether, these results suggest that restrictor adhesion formation may be associated with altered redox modulation. Similar intervention in a rat model seems to support the role of VC in tendinopathy treatment. In this context, local injection of a high dose of VC (150 mg daily) accelerates the healing of the Achilles tendon [3], indicating that local treatment with VC promotes the histological and functional recovery of ruptured Achilles tendons [55]. Moreover, other studies have shown that local inhibition of nitric oxide synthase (NOS) accelerates tendon recovery in tenotomised rats. The positive effect of NOS inhibition on tendon repair demonstrates that nitric oxide (NO) negatively regulates tissue recovery [54]. Altogether, the presented evidence suggests that VC can favor tendon recovery by acting against oxidative stress that occurs in tendon degeneration.

## Data Availability

Not applicable.

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
