# Peer review of "Effect of Vitamin C on Tendinopathy Recovery: A Scoping Review"

_nutrients, 2022, doi:10.3390/nu14132663_

Round 1

Reviewer 1 Report

This is a comprehensive article. I think it was better to explain the role of vitamin C in the introduction section. You have only described Tendinopathy. 

Author Response

REVIEWER-1

This is a comprehensive article. I think it was better to explain the role of vitamin C in the introduction section. You have only described Tendinopathy.

ANSWER: The first two paragraphs at the beginning of INTRODUCTION explain the proposed roles (antioxidant and cofactor for collagen synthesis) of vitamin C in tendinopathy treatment as suggested by the Reviewer. This change has affected the number of References that has been adapted accordingly.

Reviewer 2 Report

This study aimed to evaluate critically the efficacy of vitamin C (VC) supplementation in the treatment of tendinopathies. Major revisions are required and I suggest changing the design from a systematic to a scoping review because from a methodological perspective a lot of data are missing following the PRISMA guidelines.

The introduction paragraph and its subheadings should be revised in more proper form. Maybe it would be more fluent if the “1.2. Tendon structure and homeostasis”  follow the introduction and then put the pathogenesis (as a single subparagraph) with the three sections (Reactive Tendinopathy (stage I)- Unstructured tendon- Degenerative tendinopathy). Probably must be improved also the section related to Vitamin C, both as a supplement and as a cofactor in the collagen synthesis and the difference between ascorbic acid and ascorbate.

Methodology section

Specify the date when each source was last searched or consulted. It was registered on Prospero? Provide registration information for the review, including register name and registration number, or state that the review was not registered.

There is discordance between the figure 1 and the words used for the search reported in the material and methods. Inclusion and exclusion criteria should be extensively reported in a table: should the study must be written in English language only?

Maybe the 4 studies on surgery anastomosis could be included in the excluded ones for title ad abstract (153 excluded). Please report figure 1 in the text.

Outcomes are not well specified. Moreover, it will be specified for each outcome the effect measure(s) (e.g. risk ratio, mean difference) used in the synthesis or presentation of results.

Certainty of evidence is not reported and not discussed

For the in vitro studies excluded why do you add in vitro studies on human tissue?

The comment “In the assessment of bias (Figure 2), it is interesting to note that in the laboratory animal studies, randomisation of animals is either absent or not discussed. It is quite possible that it is not important and does not influence the study, but it is not known. On the other hand, this means that the placement in one group or another may be compromised by some of the researchers, who do not know whether or not they are aware of the therapy administered. Finally, it is common for the pathologist to work without knowing which animal the sample comes from, but it is not indicated, except in one study (58), that the samples were anonymous and randomly given to the specialist. ” should be reported in the discussion section.

Maybe it would be appropriate to report firstly both in results and in discussion the data obtained from the preclinical studies and then from the clinical ones.

Table 1 other interventions than molecules (stretching etc) should be reported in a distinct column. Timing of intervention should be reported also as frequency/week or month. More details about the participants should be added (only male? Only sport people?)

the authors reported tendon site and not the type of injury, it could be useful to know especially for the human sample the cause (sport? Work? Overuse?)

positive results reported in the impact column are those statistically significant? A p-value of referment should be added.

Table 2,  row 2,  the 2 dosages are the same. Please remove the supplementary ones.

In both the table, in the N section, sometimes two values are reported and sometimes one, without any indication about the number of control subjects, please check and change these data.

Discussion must be rewritten and improved.

In the discussion the cut “Although VC is not commonly used to treat musculoskeletal problems, is recommended to prevent the onset of limb pain syndrome after surgery, however, the mechanism is not known (60,61)” results out of context.

It is not clear “VC injection, immediately after surgical, repair prevented a decrease in GSH level at two weeks postoperatively and improved tendon release.”

Who is the subject? “Indirectly suggest that restrictor adhesion formation may be associated with altered redox modulation.”

The sentence “It is important to note that these NSAIDs may affect the healing of the tissues that make up the musculoskeletal system, and specifically the pro- liferation of tenocytes after injury (70). So much so that currently, the existing scientific references do not provide sufficient evidence for or against the use of NSAIDs after an acute injury or surgical repair of the tendon-bone interface (71–73). However, and in the belief that the contribution of the CV as a protector against the appearance of tendinopathies has been studied in vivo and in vitro (74). When NSAIDs are administered directly in the peritendinous area, it seems that the anesthetic and anti-inflammatory combination can be effective (75).” Is not clear, please rewrite and mention Is not clear, please rewrite and mention the role of hyaluronic acid in tendinopathy as a novel therapeutic strategy citing Crimaldi S, et al The Role of Hyaluronic Acid in Sport-Related Tendinopathies: A Narrative Review. Medicina (Kaunas). 2021 Oct 12;57(10):1088. doi: 10.3390/medicina57101088. PMID: 34684125; PMCID: PMC8537182.

Moreover, what about the role of physical therapies like shock wave therapies as confounding factor that further contributes the tendon healing?

English must be strongly improved and the text shows many typos.

Author Response

REVIEWER-2

This study aimed to evaluate critically the efficacy of vitamin C (VC) supplementation in the treatment of tendinopathies. Major revisions are required and I suggest changing the design from a systematic to a scoping review because from a methodological perspective a lot of data are missing following the PRISMA guidelines.

ANSWER: We agree from the Reviewer that the term “scoping review” is more appropriate. This has been changed in title accordingly.

The introduction paragraph and its subheadings should be revised in more proper form. Maybe it would be more fluent if the “1.2. Tendon structure and homeostasis” follow the introduction and then put the pathogenesis (as a single subparagraph) with the three sections (Reactive Tendinopathy (stage I)- Unstructured tendon- Degenerative tendinopathy). Probably must be improved also the section related to Vitamin C, both as a supplement and as a cofactor in the collagen synthesis and the difference between ascorbic acid and ascorbate.

ANSWER: “Tendon structure and homeostasis” is now Section 1.1 of Introduction. Therefore, “Etiopathogenesis” is Section 1.2. Coincident with Reviewer-1 suggestion, the proposed roles of Vitamin C are indicated in the first 2 paragraphs of INTRODUCTION.

Methodology section

Specify the date when each source was last searched or consulted. It was registered on Prospero? Provide registration information for the review, including register name and registration number, or state that the review was not registered.

ANSWER: Review is not registered in PROSPERO. This is indicated in the last sentence of “Materials and Methods” section.

There is discordance between the figure 1 and the words used for the search reported in the material and methods. Inclusion and exclusion criteria should be extensively reported in a table: should the study must be written in English language only?

ANSWER: Discordance between Figure 1 and text has been corrected. Inclusion criteria are indicated in the new Table 1 and exclusion criteria are indicated in Figure 1.

Maybe the 4 studies on surgery anastomosis could be included in the excluded ones for title ad abstract (153 excluded). Please report figure 1 in the text.

ANSWER: The 4 studies of surgery anastomosis have been included with the rest of excluded studies.

Outcomes are not well specified. Moreover, it will be specified for each outcome the effect measure(s) (e.g. risk ratio, mean difference) used in the synthesis or presentation of results.

Certainty of evidence is not reported and not discussed.

ANSWER: Selected publications do not give enough information to perform this analysis. In any case, we give information regarding significance in Tables 2 and 3.

For the in vitro studies excluded why do you add in vitro studies on human tissue?

ANSWER: We only have considered the study of Shaw et al (2017) because it provides key information (Table 2).

The comment “In the assessment of bias (Figure 2), it is interesting to note that in the laboratory animal studies, randomisation of animals is either absent or not discussed. It is quite possible that it is not important and does not influence the study, but it is not known. On the other hand, this means that the placement in one group or another may be compromised by some of the researchers, who do not know whether or not they are aware of the therapy administered. Finally, it is common for the pathologist to work without knowing which animal the sample comes from, but it is not indicated, except in one study (58), that the samples were anonymous and randomly given to the specialist. ” should be reported in the discussion section.

ANSWER: The paragraph has been moved to the beginning of Discussion section.

Maybe it would be appropriate to report firstly both in results and in discussion the data obtained from the preclinical studies and then from the clinical ones.

ANSWER: The studies in animals could be considered as pre-clinical. This is reported in the last sentence of the first paragraph in “Results” section.

Table 1 other interventions than molecules (stretching etc) should be reported in a distinct column. Timing of intervention should be reported also as frequency/week or month. More details about the participants should be added (only male? Only sport people?)

The authors reported tendon site and not the type of injury, it could be useful to know especially for the human sample the cause (Sport? Work? Overuse?)

Positive results reported in the impact column are those statistically significant? A p-value of referment should be added.

ANSWER: There is not enough room to place a new column. Therefore, we distributed the column indicating “Compounds/Intervention”.

Table 2, row 2, the 2 dosages are the same. Please remove the supplementary ones.

ANSWER: Dosages are different: 5 mg/mL and 50 mg/mL. However, it seems that after 2 weeks, 5 mg/mL is better than 50. This point has been indicated in the “Discussion” accordingly.

In both the table, in the N section, sometimes two values are reported and sometimes one, without any indication about the number of control subjects, please check and change these data.

ANSWER: Tables have been changed accordingly.

Discussion must be rewritten and improved.

In the discussion the cut “Although VC is not commonly used to treat musculoskeletal problems, is recommended to prevent the onset of limb pain syndrome after surgery, however, the mechanism is not known (60,61)” results out of context.

ANSWER: The paragraph has been deleted and the starting point of Discussion has been changed.

It is not clear “VC injection, immediately after surgical, repair prevented a decrease in GSH level at two weeks postoperatively and improved tendon release.”

Who is the subject? “Indirectly suggest that restrictor adhesion formation may be associated with altered redox modulation.”

ANSWER: We agree that this paragraph is confusing. We provided a new version accordingly.

The sentence “It is important to note that these NSAIDs may affect the healing of the tissues that make up the musculoskeletal system, and specifically the pro- liferation of tenocytes after injury (70). So much so that currently, the existing scientific references do not provide sufficient evidence for or against the use of NSAIDs after an acute injury or surgical repair of the tendon-bone interface (71–73). However, and in the belief that the contribution of the CV as a protector against the appearance of tendinopathies has been studied in vivo and in vitro (74). When NSAIDs are administered directly in the peritendinous area, it seems that the anesthetic and anti-inflammatory combination can be effective (75).” Is not clear, please rewrite and mention Is not clear, please rewrite and mention the role of hyaluronic acid in tendinopathy as a novel therapeutic strategy citing Crimaldi S, et al The Role of Hyaluronic Acid in Sport-Related Tendinopathies: A Narrative Review. Medicina (Kaunas). 2021 Oct 12;57(10):1088. doi: 10.3390/medicina57101088. PMID: 34684125; PMCID: PMC8537182.

ANSWER: Paragraph is confusing. We have written a new version accordingly

Moreover, what about the role of physical therapies like shock wave therapies as confounding factor that further contributes the tendon healing?

ANSWER: This is not the goal of the present manuscript, but this could be a subject for a future review.

English must be strongly improved and the text shows many typos.

ANSWER: The whole manuscript has been revised accordingly.

Reviewer 3 Report

I found this study to be very informative and well organized.

I have only one point to make.

Was there any study that examined the combined effects of vitaminâ…­ and exercises therapy?

This study focused only on vitaminâ…­, but I think nutrition and exercise need to be considered as a set.

Author Response

REVIEWER-3

I found this study to be very informative and well organized.

I have only one point to make.

Was there any study that examined the combined effects of vitamin â…­ and exercises therapy?

ANSWER: Studies of this type combining exercise together with vitamin C administration are indicated in Table 2.

This study focused only on vitamin â…­, but I think nutrition and exercise need to be considered as a set.

ANSWER: Since “nutrition and exercise” would give references for a great review, we believe that this assessment has to be the base for all sport practitioners. We think that this sentence has to be last sentence of the review, indicating to readers the importance of “nutrition and exercise”.

Reviewer 4 Report

The study entitled “Effect of Vitamin C (VC) on Tendinopathy Recovery. A systematic Review” summarizes the effect of vitamin C on the recovery of tendinopathies. Structured search with appropriate keyword was carried out in various databases including SCOPUS, Web of Science and Medline (PubMed). The inclusion and exclusion criteria are well defined and is suitable for this study. This structures review showed that vitamin C supplementation increased the production of collagen and subsequently improved the tendinopathies. Moreover, this study clearly indicates that the deficiency in vitamin C is mainly associated with reduced procollagen synthesis, and hydroxylation of proline and lysine residues leading to sluggish tendon repairment.

However, few queries need to be answered:

1.      Technical jargons, typos, mistakes in spelling and punctuation throughout the manuscript

2.      The number of invitro studies, reviews, erratum, and letter to the editor which are excluded must be mentioned clearly (separately)

Author Response

REVIEWER-4

The study entitled “Effect of Vitamin C (VC) on Tendinopathy Recovery. A systematic Review” summarizes the effect of vitamin C on the recovery of tendinopathies. Structured search with appropriate keyword was carried out in various databases including SCOPUS, Web of Science and Medline (PubMed). The inclusion and exclusion criteria are well defined and is suitable for this study. This structured review showed that vitamin C supplementation increased the production of collagen and subsequently improved the tendinopathies. Moreover, this study clearly indicates that the deficiency in vitamin C is mainly associated with reduced procollagen synthesis, and hydroxylation of proline and lysine residues leading to sluggish tendon repairment.

However, few queries need to be answered:

  1. Technical jargons, typos, mistakes in spelling and punctuation throughout the manuscript.

ANSWER: This concern is coincident with the last concern raised by Reviewer-2. In this line, the whole manuscript has been revised accordingly.

  1. The number of invitro studies, reviews, erratum, and letter to the editor which are excluded must be mentioned clearly (separately).

ANSWER: See new version of Figure 1.

Round 2

Reviewer 2 Report

The manuscript seems improved and all the corrections suggested have been added. Given the lack of some steps required for a systematic review, it was suggested to modify it in scoping review. This change with the description of the new kind of review should be mentioned within the methods section, which should be rewritten in light of this variation.

Author Response

REVIEWER-2-2

The manuscript seems improved and all the corrections suggested have been added. Given the lack of some steps required for a systematic review, it was suggested to modify it in scoping review. This change with the description of the new kind of review should be mentioned within the methods section, which should be rewritten in light of this variation.

ANSWER: We agree with the Reviewer that the manuscript is a “scoping review”, trying to show the key aspects of vitamin C in tendinopathy treatment and offering new lines for future research. This aspect is indicated in the title as well as in the first paragraph of “Materials and Methods” section as suggested.